3D reconstruction of toys based on adaptive scaled neural radiation field

Zou Jiajun 1
Liu Shaojiang 1
Wang Feng 1
Ni Weichuan 2
Ye Shitong 2 yeshitong@gdhsc.edu.cn
1 Department of Information and Intelligence Engineering, Guangzhou Xinhua University , Dongguan, Guangdong , China
2 Department of Artificial Intelligence, Guangzhou Huashang College , Guangzhou, Guangdong , China
Asif Muhammad
Electronic publication date: 2025 Jul 29
Publication date: 2025
Volume: 11
Electronic Location ID: e3053
Received 2025 Apr 29; Accepted 2025 Jun 30
Copyright: © 2025 Zou et al.
Copyright year: 2025
Copyright holder: Zou et al.
License: This is an open access article distributed under the terms of the Creative Commons Attribution License, which permits unrestricted use, distribution, reproduction and adaptation in any medium and for any purpose provided that it is properly attributed. For attribution, the original author(s), title, publication source (PeerJ Computer Science) and either DOI or URL of the article must be cited.
License URL: https://creativecommons.org/licenses/by/4.0/

Keywords: Three-dimensional reconstruction, Neural radiation field, Multi-task learning, Adaptive scale adjustment, Conditional diffusion modeling

Funding: Guangdong Province Key Construction Discipline Research Capacity Enhancement 2021ZDJS144, 2024ZDJS130 Key Research Platform of Guangdong Ordinary Colleges and Universities 2024ZDZX3035 Characteristic Innovation Category Project of Guangdong Ordinary Colleges and Universities 2024KTSCX127 School-level Scientific Research Project of Guangzhou Xinhua University 2024KYCXTD02 Young Innovative Talents Category Project of Guangdong Ordinary Colleges and Universities 2023KQNCX124, 2024KQNCX076 This research was supported by the Guangdong Province Key Construction Discipline Research Capacity Enhancement Project (Nos. 2021ZDJS144, 2024ZDJS130), the Key Research Platform of Guangdong Ordinary Colleges and Universities and Project No. 2024ZDZX3035, the Characteristic Innovation Category Project of Guangdong Ordinary Colleges and Universities (No. 2024KTSCX127), the School-level Scientific Research Project of Guangzhou Xinhua University (No. 2024KYCXTD02), the Young Innovative Talents Category Project of Guangdong Ordinary Colleges and Universities (Nos. 2023KQNCX124, 2024KQNCX076). The funders had no role in study design, data collection and analysis, decision to publish, or preparation of the manuscript.

==============================
With the rapid development of computer vision technology, 3D reconstruction of toys under single-view conditions still faces significant challenges in terms of detail loss and color distortion. For this reason, this article proposes an adaptive scale neural radiance fields (AS-NeRF) model to enhance the accuracy and realism of 3D toy reconstruction. The method constructs a multi-task feature extraction network based on the Vision Transformer, which simultaneously extracts and fuses multidimensional features such as texture, shape, color, and depth through a task dynamic modulation mechanism and a dynamic adapter layer, providing a rich and accurate contextual feature representation. The NeRF model is enhanced to incorporate an adaptive scaling mechanism that dynamically optimizes rendering sampling accuracy according to the local complexity of the scene. Spectral sensing techniques are integrated to reproduce the true colors of materials accurately. Finally, the conditional diffusion model is deeply integrated with NeRF, and high-dimensional conditional vectors are used to guide the inverse diffusion process in generating unobserved images with consistent geometric structure and physical properties. Experiments on the Co3D toy dataset demonstrate that AS-NeRF significantly outperforms existing mainstream methods in terms of peak signal-to-noise ratio (PSNR), structural similarity (SSIM), loss of perceptions (LPIPS), and Chamfer distance, thereby verifying the validity and advantages of the proposed method for high-quality toy 3D reconstruction tasks.

Introduction

With the rapid development of computer vision and graphics, 3D reconstruction technology plays a crucial role in various fields, including virtual reality, augmented reality, autonomous driving, and robot navigation, among others. High-quality, real-time, and accurate 3D reconstruction methods can not only enhance the user experience but also provide solid technical support for related applications. However, traditional 3D reconstruction methods still face many challenges in handling complex scenes, fusing information from multiple viewpoints, and restoring details (Aharchi & Ait Kbir, 2020; Anciukevičius et al., 2023; Sun et al., 2021; Alldieck, Zanfir & Sminchisescu, 2022).

Among the many application scenarios, 3D reconstruction of toys is gradually gaining attention as a unique and essential research direction. As an integral part of children’s entertainment and education, the diverse shapes, colors, and detailed features of toys make 3D reconstruction technology a versatile tool with a wide range of potential applications in toy design, virtual display, online shopping, and collection management. For example, toy designers can utilize high-precision 3D models for design iteration and prototype testing. At the same time, consumers can preview the appearance and details of toys before purchasing them through virtual reality technology. Additionally, toy collectors can digitize and preserve valuable collectibles using 3D reconstruction technology, thereby preventing damage and loss of physical toys. However, the 3D reconstruction of toys also faces unique challenges, such as the high complexity of detailed structures, diverse materials and colors, as well as the requirement for real-time and high accuracy.

Traditional toy 3D reconstruction methods, such as stereo vision, structured light, and laser scanning, although excellent in some applications, have several limitations, including the inability to process complex details, fuse information from multiple viewpoints, and operate in real-time, as well as high costs (Tian et al., 2022; Wang et al., 2020; Pintore et al., 2020). With the rise of deep learning technology, neural radiance fields (NeRF) has gradually become a research hotspot as an emerging neural network-based 3D reconstruction method, which achieves high-quality viewpoint synthesis and detail restoration by implicitly representing the scene’s bulk density and color (Bian et al., 2023; Hu et al., 2023; Chen & Lee, 2023). By implicitly representing the scene’s bulk density and color, NeRF can realize high-quality viewpoint synthesis and detailed reconstruction of toys. The core idea is to use a multilayer perceptron network to predict the color and density of each point in 3D space based on the input 2D image viewpoint information, thereby rendering an image from any viewpoint. Although NeRF has made significant progress in 3D reconstructing static scenes, it still faces several limitations in practical applications. For example, NeRF often faces challenges such as computational inefficiency and insufficient reconstruction accuracy when handling dynamic scenes, limited viewpoint data, and high-resolution reconstruction (Yan, Li & Lee, 2023; Cui et al., 2024; Huang et al., 2023). In addition, NeRF relies heavily on the number of input viewpoints, and the reconstruction results under conditions of a single view or a small number of viewpoints are often unsatisfactory.

Existing 3D reconstruction methods often face outstanding challenges, such as insufficient accuracy and limited reconstruction quality, when dealing with the complexities of detailed objects, diverse materials, rich colors, and the limited number of viewpoints unique to toy scenes. In particular, the standard NeRF method is highly dependent on a large number of viewpoint inputs, which makes it challenging to effectively address the issues of missing details and color distortion under a single or a small number of viewpoints. Therefore, there is an urgent need to develop a new method that can adapt to sparse viewpoint scenarios while accurately restoring rich details and true colors of toys. To address the above research gaps, this study proposes an AS-NeRF model that accurately captures texture, shape, color, and depth information by designing a multi-task learning feature extraction network, MTL-FeatureNet. Adaptive scale adjustment and spectral sensing techniques are introduced to dynamically optimize the rendering accuracy of local regions and improve the accuracy of material color reproduction. We innovatively integrate the conditional diffusion model with NeRF, utilizing high-dimensional conditional coding to guide the inverse diffusion generation process and efficiently produce unobserved viewpoint images with consistent geometric and physical attributes. This approach overcomes the limitations of existing methods, which rely on sparse viewpoints and suffer from insufficient detail reproduction and color distortion. The main contributions of AS-NeRF include: (1) Multi-task learning feature extraction: By designing a multi-task learning feature network, AS-NeRF can simultaneously extract multidimensional features, such as texture, shape, color, and depth, which provides rich contextual information and enhances the ability to detail restoration in the reconstruction process. This is especially important for objects with rich and diverse details, such as toys.

(2) Adaptive scale adjustment and spectral awareness: An adaptive scale adjustment mechanism is introduced to dynamically optimize the rendering accuracy according to the complexity of the toy’s shape and details, and spectral awareness technology is combined to improve the realism of the color reproduction and the detail performance, ensuring that the details can be accurately reconstructed at different scales.

(3) Effective integration of conditional diffusion model: The conditional diffusion model is organically combined with the feature extraction network and NeRF model to guide the inverse diffusion process through the high-dimensional conditional vectors, which ensures that the generated images are accurate and coherent in terms of geometrical and physical attributes and that the high-quality reconstruction effect can still be maintained, especially in the case where the number of viewpoints is small. This is especially important for toy models that require reconstruction with complex details from a limited number of views.

This thesis is organized as follows: “Related Work” primarily reviews the research progress in the field related to this study, focusing on NeRF and its improved methods, particularly the latest results in 3D reconstruction and multiview information fusion. “Model Architecture and Feature Extraction Network” provides a detailed description of the overall model architecture of AS-NeRF, with a focus on the design and implementation of the Multi-Task Learning Feature Extraction Network (MTL-FeatureNet). “Improved NeRF Model: Combining Adaptive Scaling with Spectral Sensing” describes the improved NeRF model in AS-NeRF, focusing on the adaptive scale adjustment mechanism and the integration method of spectral perception techniques. “Conditional Diffusion Model” introduces the application of the conditional diffusion model in AS-NeRF, explaining how it is combined with the feature extraction network and the NeRF model. “Experimental Analysis” presents a comprehensive experimental evaluation of the method in this article, comparing it with a standard procedure on the Co3D dataset and providing comparative results on several evaluation metrics. “Conclusion” summarizes the full article, outlining the main contributions and research results of AS-NeRF in the 3D reconstruction of toys. The limitations of the research are discussed, and future research directions and possible improvements are proposed.

Related work

3D reconstruction techniques have been extensively researched and applied in the fields of computer vision and graphics, particularly in neural network-based methods. NeRF, as an emerging 3D reconstruction method, achieves high-quality viewpoint synthesis and detail restoration by implicitly representing the scene’s bulk density and color (Mildenhall et al., 2021). Since the proposal of NeRF, numerous improvements and extensions have emerged from academia and industry to address the limitations of NeRF in various application scenarios and enhance its performance in complex environments. With the successful application of Transformers in the field of computer vision, researchers have attempted to integrate them into the NeRF model to enhance its feature extraction capabilities and viewpoint synthesis performance. Lin et al. (2023) proposed a visual Transformer-based NeRF approach, which enhances the NeRF under single-input image conditions by introducing the Transformer architecture to the viewpoint synthesis capability of NeRF. However, such methods primarily focus on feature extraction and structure optimization, lacking further modeling of local details and geometric consistency, which makes it challenging to meet the demand for high-precision reconstruction of objects with complex information, such as toys. SparseFusion enhances 3D reconstruction under limited viewpoint conditions by fusing sparse viewpoint information and utilizing a diffusion model of the viewpoint conditions (Zhou & Tulsiani, 2023). This method enhances the reconstruction capability of NeRF under sparse viewpoint data by distilling viewpoint information and reducing its dependence on a large number of input images. However, it still has limitations in fine geometry and realistic color reproduction, especially underperforming in toy scenes with complex materials and rich colors.Mip-NeRF introduces a multiscale representation to reduce aliasing effects, thereby enhancing the rendering quality of NeRF at high resolutions (Barron et al., 2021). Although the method improves the fineness of the reconstructed images, it lacks flexibility in dynamically adjusting the sampling accuracy, leading to a significant increase in computational cost.

In 3D toy reconstruction, it typically faces the limitation of single-view or few-view data. PixelNeRF enhances the reconstruction capability of NeRF with few-view data through pixel-level feature extraction (Yu et al., 2021). To enhance the geometric consistency of 3D reconstruction, researchers have combined neural implicit surface methods with NeRF. For example, Geo-Neus improves the accuracy and detail restoration of multiview reconstruction by combining geometrically consistent neural implicit surface learning (Fu et al., 2022). Neus, proposed by Wang et al. (2021), further improves the geometrical consistency and detail restoration of multiview reconstruction by combining neural implicit surface learning and volumetric rendering techniques. Ran et al. (2023) propose the Neurar model, which enhances the robustness and accuracy of NeRF in autonomous 3D reconstruction by introducing neural uncertainty. This approach not only improves the model’s generalization ability but also performs well in handling uncertain data. In addition, Unisurf, proposed by Oechsle, Peng & Geiger (2021), unifies the neural implicit surface with the radiation field, further improving the performance of NeRF in complex scenes. High-fidelity 3D reconstruction is crucial for applications involving detail-rich objects, such as toys. Neuralangelo significantly enhances the ability of NeRF in detail restoration and high-precision reconstruction through high-fidelity neural surface reconstruction (Li et al., 2023). The single-view 3D scene reconstruction method proposed by Chen et al. (2024) further enhances NeRF through high-fidelity shape and texture reconstruction, enabling detailed restoration under single-view conditions. Neus2, proposed by Wang et al. (2023), improves the efficiency and accuracy of NeRF in multiview reconstruction by rapidly learning neural implicit surfaces.

Model architecture and feature extraction network

Model architecture

The adaptive scale neural radiance fields (AS-NeRF) model proposed in this study aims to comprehensively improve the accuracy and quality of 3D reconstruction of toys by integrating multi-task learning feature extraction, adaptive scale adjustment with spectral sensing, and conditional diffusion model. The overall architecture is shown in Fig. 1. AS-NeRF consists of three main components, each of which plays a key role in the overall reconstruction process and enables efficient 3D reconstruction through close collaboration.

Figure 1 Overall architecture of the AS-NeRF approach.

The multi-task learning feature extraction network (MTL-FeatureNet) is responsible for extracting multidimensional features, including texture, shape, color, and depth, from the input single 2D toy view. These rich features provide the necessary contextual information for subsequent 3D reconstruction. MTL-FeatureNet is based on an improved Vision Transformer (ViT) architecture, which optimizes the feature extraction process by introducing a task-dynamic modulation layer, allowing it to better adapt to multi-task learning environments and enhancing the ability to extract high-quality features from a single view.

The extracted multidimensional features are then passed to an improved NeRF model that integrates an adaptive scaling mechanism and spectral sensing techniques. The adaptive scaling mechanism dynamically optimizes the rendering accuracy according to the complexity of toy shapes and details, ensuring accurate reconstruction of details at different scales, thus improving the accuracy and computational efficiency of 3D reconstruction. Spectral awareness technology introduces a spectral decoder, which uses multi-spectral information to enhance the color realism and material details, so that the generated 3D model has higher realism and details under various lighting conditions.

To further enhance the quality and consistency of the generated images, AS-NeRF integrates a conditional diffusion model. The model utilizes multidimensional features from MTL-FeatureNet and the improved NeRF model as conditional information to guide the backward diffusion process, ensuring consistency and accuracy of the generated viewpoint images in terms of geometric and physical properties. The conditional diffusion model not only optimizes the details of image generation, but also enhances the reconstruction ability of the model under few viewpoint conditions, enabling the generation of high-quality 3D models even with limited viewpoint data.

Figure 1 shows the overall architecture of the AS-NeRF method. In the AS-NeRF framework, a single-view image is extracted by MTL-FeatureNet to produce a local feature map and global semantic vectors. The local feature maps are sent to NeRF with ray sampling to guide the density and color prediction of each 3D sampling point. The global vectors are involved in adaptive scale selection and spectral decoding in NeRF, and on the other hand, they are compressed into conditional vectors and input into the conditional diffusion model, which are used to constrain the diffusion process and synthesize the unobserved viewpoint images quickly. In the training stage, the new viewpoints generated by the diffusion model are projected back to NeRF as pseudo-supervision, while in the inference stage, only one feature extraction is needed to render the real viewpoints of NeRF and synthesize the arbitrary viewpoints of the diffusion model at the same time, and the two paths share the features without blocking each other, which ensures that geometrically and spectrally consistent and high-quality 3D reconstruction results can be obtained under the condition of sparse viewpoints.

Feature extraction network based on multi-task learning approach

In this section, we describe how to extract multilevel features such as texture, shape, color, and depth from a single 2D toy view by MTL-FeatureNet to provide the necessary information for complex 3D reconstruction tasks.

MTL-FeatureNet is mainly based on the ViT architecture (Han et al., 2022; Liu et al., 2021; Arnab et al., 2021), but in MTL-FeatureNet we improve the traditional multi-head self-attention mechanism to optimize the extraction of features from a single 2D toy view, especially in a multi-task learning environment. The improvements include the introduction of a task dynamic modulation layer, which is a design that adjusts the self-attention weights according to the task demands. The purpose of the task dynamic modulation layer is to dynamically adjust the attention allocation according to the specific demands of different tasks. This layer optimizes the weight distribution of each head in the attention mechanism by analyzing task-related contextual information. The specific implementation is as follows.

We assume that each task is computed by a neural network module to obtain a task relevance score TRi, which is used to reflect the importance of the current feature for a particular task:

(1) TRi=MLPtask(Ci).

MLPtask refers to a multilayer perceptron that receives high-level features from a specific part of the image and outputs a real value that indicates the importance of the current feature for performing a specific task. Ci refers to regional or global features in the image after preprocessing (e.g., feature extraction network layer processing).

The above scores are then used to adjust the self-attention weights so that the model can focus on the most relevant information according to the specific needs of each task:

(2) Ati(Q,K,V)=softmax(QKTdk⋅exp(TRi))V.

Based on the task relevance score TRi, we act on the attention weights in the form of exp(TRi) to enhance or diminish the influence of specific parts to ensure that the model’s attention is more in line with the current task.

Since positional information is extremely important for understanding image content, we adopt an improved positional coding method, which not only considers pixel positions, but also incorporates task-related dynamic adjustments:

(3) PE(pos,2i)=sin(pos/10,0002i/dmodel)⋅TR,

(4) PE(pos,2i+1)=cos(pos/10,0002i/dmodel)⋅TR.

Since TR is a task relevance score from the task dynamic modulation layer, by multiplying it with the conventional computation of position encoding, a way of dynamically modulating the position encoding can be realized in such a way that it reflects the specific needs of different tasks. With this enhanced multi-head self-attention mechanism and the task dynamic modulation layer, MTL-FeatureNet is able to handle the demands of multi-task learning more efficiently by improving the accuracy and relevance of features extracted from a single 2D toy view.

For the extracted features, we also introduce feature fusion and dynamic adapter layer (DAL) in the MTL-FeatureNet architecture, which aims to further optimize and synthesize these features to ensure that they can be adapted to multiple different task requirements. The DAL works after the multi-head self-attention mechanism, receiving the feature Fglobal output from the self-attention layer and transforming it with a set of specially designed parameters to better serve a particular task:

(5) Ftask=σ(γtask⋅Wtask⋅Fglobal+btask).

In this formulation γtask is a scaling factor customized for each task, which is automatically adjusted through the learning process to optimize the transformation efficiency of the features, Wtask is the task-specific weight matrix, and btask is the bias term. This setup allows the network to not only respond to different task requirements, but also maintain its versatility and adaptability in handling various visual tasks.

To further enhance the network’s ability to handle multi-tasks, we design a composite loss function that not only considers the performance of individual tasks, but also reinforces the synergistic effect between tasks through cross-task consistency loss:

(6) L=∑k⁡λk⋅Lk+μ⋅Lcross(Ftask1,Ftask2,…).

Lcross denotes the loss of cross-task consistency, which facilitates knowledge sharing and complementarity between tasks by comparing similarities or differences between feature representations generated by different tasks:

(7) Lcross(Ftask1,Ftask2,…)=∑i≠j⁡vij⋅‖Ftaski−Ftaskj‖2

where vij is a dynamically adjusted weight that adjusts the effect of feature differences between each pair of tasks according to the correlation between different tasks. The schematic diagram of the feature extraction process for multi-task learning network is shown in Fig. 2.

Figure 2 Feature extraction process of multi-task learning network.

The pseudo-code of the algorithm for MTL-FeatureNet is shown below.

Algorithm 1 Simplified MTL-FeatureNet Pseudo-code.

Input: 2D toy image I, task set {T1, T2, …, Tn}	
Output: Task-specific features {F_task1, F_task2, …, F_taskn}	
Step 1: Feature Extraction and Task Relevance	
for each task Ti:	
    Extract initial features Ci from I	
    Compute task relevance TRi = MLP_task(Ci)	
Step 2: Task Dynamic Attention Modulation	
for each task Ti:	
    Adjust attention:	
        Ati(Q, K, V) = softmax((QK^T/sqrt(dk)) * exp(TRi)) * V	
Step 3: Dynamic Adapter Layer (DAL)	
for each task Ti:	
    Compute task-specific features:	
        F_taski = σ(γ_task ⋅ W_task ⋅ F_global + b_task)	
Step 4: Multi-Task Loss	
Define loss:	
    L = Σk λk ⋅ Lk + μ ⋅ L_cross	
Step 5: Output Features	
Return {F_task1, F_task2, …, F_taskn}	

Improved NeRF model: combining adaptive scaling with spectral sensing

Among modern 3D reconstruction techniques, NeRF has demonstrated its capability in generating high-quality detail-rich 3D visual content (Low & Lee, 2023; Yan, Li & Lee, 2023). To further enhance its performance and address the challenges in dealing with high complexity scenes and complex lighting conditions, we introduce an improved NeRF model, a model that significantly enhances the rendering accuracy and color reproduction fidelity of toy 3D models by combining techniques of adaptive scale adjustment and spectral perception.

Adaptive scaling

The next step in extracting key visual features from a single 2D toy view via MTL-FeatureNet is to utilize these features for efficient 3D reconstruction. In this section, we introduce the adaptive-scale NeRF technique, an improvement on the standard NeRF model designed to increase the accuracy and efficiency of 3D reconstruction, which is particularly suitable for processing features obtained from MTL-FeatureNet.

NeRF achieves impressive 3D reconstruction results by modeling the scene’s continuous bulk density and color in the line-of-sight direction, using multiple stacked fully connected networks (MLPs) to predict the bulk density and color. However, standard NeRFs often require a large amount of training time and computational resources when dealing with complex scenes, and do not capture enough details at different scales. Adaptive Scale NeRF allows the model to dynamically adjust the accuracy of rendering according to the complexity of the scene and the detail requirements of a specific region by introducing a scale-adaptive mechanism. This improvement is mainly to optimize the rendering efficiency and increase the model’s ability to capture details, especially when combined with advanced features from MTL-FeatureNet.

We use MTL-FeatureNet to extract image global features G. These features include not only texture, shape, and color, but also depth and illumination information through specific tasks. In order to dynamically adjust the rendering accuracy according to the extracted features during the rendering process, we define a composite adaptive scaling function:

(8) S(FA(G,x(t)),G)=sigmoid(β⋅Dcomplex(CT(FA(G,x(t)),G))).

β is a learnable scaling factor that regulates the sensitivity of the scale-adaptive function, allowing the model to find a balance between preserving detail and computational efficiency. Dcomplex denotes the deep neural network used to analyze the integrated local and global features, whose output determines the degree of scale tuning. CT(⋅) is a co-feature merging function that synthesizes the local feature Q and the global feature G in order to generate a comprehensive scene description. FA(G,x(t)) is an estimation function that determines the local features at a 3D location x(t) based on that location and the global feature G.

Combined with an adaptive scale function, this allows the NeRF rendering process to adaptively adjust the contribution of each sample point:

(9) C(r)=∫tntf⁡T(t)⋅σ(p(x(t)))⋅c(x(t),S(FA(G,x(t)),G))dt.

c(x(t),S(FA(G,x(t)),G)) is the adjusted color output, where the adaptive scale function S dynamically adjusts the color rendering at the location based on the global and estimated local features.

Spectrum-aware rendering

Spectrally aware rendering techniques (Zhang et al., 2024; Afanasyev, Voloboy & Ignatenko, 2015; Meng & Yuan, 2021) play a key role in further enhancing the realism and detailed representation of toy 3D models. This technique utilizes data acquired from multispectral images based not only on conventional red, green, blue (RGB) information, but also on a wider range of spectral information to enhance the color realism and detailed representation of materials in toy 3D models. Spectrally-aware rendering is able to simulate the physical interactions of light at different wavelengths, thus providing more accurate color reproduction under various lighting conditions, especially at the edges of the spectrum and on special materials. In this article, we model the spectrally-aware NeRF by introducing a spectral decoder that not only responds to spectra in the visible range, but also processes spectral information beyond this range. This allows the model to take into account more complex spectral data, such as infrared and ultraviolet information, in the rendering process, thus improving the color fidelity and visual effects of the scene. The spectral decoder we used is formulated as follows:

(10) c(x,ε)=softmax(∑n=1N⁡Gn⋅e−(ε−μn)22δn2)⋅MLPcolor(F(x,ε)).

c(x,ε) denotes the color output at position x and optical wavelength ε. μn,δn2,Gn are the center wavelength, variance and weight of the n-th spectral component, respectively, which help the model to precisely control the spectral response at different wavelengths. The MLPcolor is a specially designed multilayer perceptron for decoding the color based on the spectral features F(x,ε), which takes into account the effects from different wavelengths.

Combined with the spectral perception technique, the rendering formula of NeRF is further improved as:

(11) C(r)=∫tntf⁡T(t)⋅σ(p(x(t),ε))⋅c(x(t),ε,S(FA(G,x(t)),G))dt.

p(⋅),c(⋅) are the bulk density and color, respectively, which are now dependent on the wavelength of light ε. This allows the model to dynamically adjust the rendering under different lighting conditions, which not only takes into account the effect of the wavelength ε on the color and the bulk density, but also integrates an adaptive scale adjustment function S(FA(G,x(t)),G) to optimize the rendering efficiency and quality. With this approach, the NeRF model is not only able to adjust the rendering accuracy to adapt to the scene complexity, but also able to reproduce the colors more realistically under various lighting conditions. With this approach of combining adaptive scaling and spectral perception techniques into NeRF models, we can realize higher quality 3D visual content. The implementation process of the NeRF method combining adaptive scaling and spectral perception is shown in Fig. 3.

Figure 3 NeRF combining adaptive scaling with spectral perception.

The pseudo-code of the algorithm that improves on the NeRF method is shown below.

Algorithm 2 Adaptive-scale and spectral-aware NeRF Pseudo-code.

Input:	
    2D toy image I, camera parameters P,	
    global features G from MTL-FeatureNet,	
    spectral components ε	
Output:	
    Rendered 3D image C(r)	
Step 1: Initialize NeRF Model	
    Load MTL-FeatureNet to extract global features G from input image I	
    Define volume density network σ(x)	
    Define color network c(x, ε, S)	
Step 2: Adaptive-Scale Function	
    Extract local features Q from 3D position x(t) using:	
        FA(G, x(t)) = Local feature estimation based on global features and position	
    Compute adaptive scale S(FA(G, x(t)), G):	
        S(FA(G, x(t)), G) = σ (β ⋅ D_complex (CT (FA(G, x(t)), G ) ) )	
    Where:	
        D_complex is a depth network	
        CT( ⋅) is a feature combination function for local and global features	
Step 3: Spectral-Aware Color Estimation	
    Estimate color c(x(t), ε) using spectral decoder:	
        c(x(t), ε) = softmax( Σ_{n=1}^N G_n ⋅ exp(-(ε - μ_n)^2/(2 δ_n^2)) ) ⋅	
                    MLP_color( F(x(t), ε) )	
    Where:	
        μ_n = center wavelength of the n-th spectral component	
        δ_n^2 = variance	
        G_n = weight	
Step 4: Volume Rendering with Adaptive Scale and Spectral Color	
    For each ray r, compute the rendered color C(r):	
        C(r) = ∫_{t_n}^{t_f} T(t) ⋅ σ(x(t), ε) ⋅	
               c(x(t), ε, S(FA(G, x(t)), G)) dt	
    Where:	
        T(t) = accumulated transmittance	
        σ(x(t), ε) = spectral-aware volume density	
Step 5: Output Rendered Image	
    Return the final rendered image C(r) considering adaptive scale and spectral awareness	

Conditional diffusion model

In this study, the central role of the conditional diffusion model (Yu et al., 2023; Zhang, Rao & Agrawala, 2023; Zhu et al., 2023) is to generate high-quality viewpoint images by utilizing the multidimensional features provided by MTL-FeatureNet and the viewpoint data generated by the improved NeRF model. By combining these advanced models with the conditional diffusion model, we are able to efficiently generate images of unobserved viewpoints that are consistent with geometric and physical realism. The features extracted via MTL-FeatureNet are used as inputs to the conditional diffusion model, which utilizes these features to guide the generation process, ensuring that the newly generated images are not only visually coherent, but also consistent with the original viewpoints in terms of geometric and physical properties. In order to effectively utilize these features, we have improved the traditional backward diffusion process, especially in the fusion and use of conditional information:

(12) ρθ(Ot−1|Ot,c)=N(Ot−1;μθ(Ot,c,t),∑θ⁡(Ot,c,t))

where μθ and Σθ are conditionally dependent parameters that are not only adjusted according to the current noisy image Ot, but also depend on the composite feature c extracted from the MTL-FeatureNet and NeRF models. This design allows the model to more accurately recover the details of the target image during the denoising process, especially when dealing with complex illumination and depth information. The composite feature c is obtained from the encoder as a high-dimensional conditional vector:

(13) c=EC(G^,J).

Global features G^ include NeRF-generated 3D scene information such as viewpoint-dependent scene geometry and illumination patterns. J denotes local features from MTL-FeatureNet, including detail information such as texture and shape. These features are synthesized in the encoder to generate conditional vectors that can guide image generation.

The training of the conditional diffusion model involves optimizing a composite loss function that is designed to improve the quality of the generated images and to ensure that the conditional information is used efficiently:

The training of the conditional diffusion model involves optimizing a composite loss function that is designed to improve the quality of the generated images and to ensure that the conditional information is used efficiently:

(14) T(θ)=Eq(Ot|O0)[logq(Ot−1|Ot,O0)ρθ(Ot−1|Ot,c)].

This loss function helps the model learn exactly how to reconstruct clear and accurate images of new viewpoints in reverse, guided by added noise and conditional information. The improved conditional diffusion model improves the ability to generate 3D models from a limited number of viewpoints, and by combining information from different data sources, it enhances the accuracy and realism of the model in generating images from unobserved viewpoints.

Experimental analysis

Experimental environment

For the hardware of this experiment, a server equipped with NVIDIA GeForce RTX 4090 GPU with 24GB of video memory was used, the processor was an Intel Core i9-10900K with 10 cores and 20 threads, paired with 64GB of DDR4 RAM, and the storage was equipped with a 2TB NVMe SSD. For the software environment, Ubuntu 20.04 LTS operating system was used, Python 3.8 was chosen as the programming language, and PyTorch 1.8 was used as the deep learning framework. The experimental dataset is the Co3D Toys dataset, which is divided into 80% for the training set and 20% for the testing set. The Co3D Toys subset is derived from the Canonical 3D dataset published by Meta AI. The subset contains 51 types of toy objects, 1,645 individual scenes, and about 88,000 images. The original resolution is 960 × 540 px, and in order to ensure that the algorithm can adapt to the high resolution, we randomly crop and scale the images to 512 × 512 px in the training phase, and keep 768 × 768 px in the testing phase. Co3D provides synchronized internal and external parametric calibration, and the camera trajectory is distributed as an ellipsoidal cone around a single object: elevation angle 10°–70°, and azimuth angle 0°–360°, We follow the official few view classification, where 80% of the scenes (≈1,316 scenes) in each category are used as the training set, and 20% of the scenes (≈329 scenes) are used as the test set, and 1–5 views are uniformly sampled by trajectory as inputs, and the rest of the views are used as inputs. Five views as inputs and the rest of the views are used for evaluation to ensure the comparability of the methods under sparse view and multiview conditions.

Experimental program

In order to comprehensively evaluate the performance of the AS-NeRF method in 3D reconstruction tasks, we carried out detailed experimental parameter settings for the key components of this method: the number of layers of the Transformer encoder was set to 12, the embedding dimensions were set to 768 dimensions, and 12 attentional heads were used, each with a dimensionality of 64, for each task (texture, shape, color, and depth), we introduced an independent task relevance scoring module, whose multilayer perceptron contains two layers, the hidden layer size is 256, and the activation function adopts ReLU. The initial value of the scaling factor γtask is set to 1.0, which is automatically learned and optimized through the training process. λk is set for each task, and the weights of texture, shape, color, and depth are 1.0, 1.0, 0.8, and 1.0, respectively. μ is set to 0.1, in order to balance the individual task performance and inter-task synergy. vij is dynamically adjusted according to task relevance, with an initial value of 0.5. β is initialized to 1.0 and automatically adjusted by backpropagation during the training process to optimize the sensitivity of the scale-adaptive function. N is set to 31, which covers major wavelengths in the range of 400 to 700 nm, with one component at 10 nm intervals. μn is set to 1.0 for the range from 400 to 700 nm at 10 nm intervals. δn2 is uniformly set to 15 to ensure the smoothness of the spectral response. Gn is initialized to 1.0 to learn the contribution of each wavelength through the training process.

In order to comprehensively evaluate the performance of AS-NeRF, we select two comparison methods, SparseFusion and PixelNeRF, respectively. PixelNeRF is typical in single-view or few-view conditions, while SparseFusion combines view-conditional diffusion and distillation strategies, which is currently recognized as an important benchmark for sparse-view reconstruction performance. Both of them cover the feature reprojection route and the diffusion-NeRF route respectively, which are the closest to the technical path of the sparse view toy reconstruction task focused in this article. The experimental standard evaluation metrics cover a variety of aspects such as reconstruction accuracy, visual quality and geometric consistency. The specific evaluation indexes are as follows: (1) Peak signal-to-noise ratio (PSNR): PSNR is used to measure the similarity between the reconstructed image and the real image at the pixel level. Its calculation formula is: (15) PSNR=10⋅log10(MAX2MSE).

MAX is the maximum pixel value of the image and MSE is the mean square error. Higher PSNR value indicates better quality of the reconstructed image.

(2) Structural similarity index (SSIM): the SSIM is used to evaluate the similarity between the reconstructed image and the real image in terms of structure, brightness and contrast, which is calculated by the formula: (16) SSIM(x,y)=(2μxμy+C1)(2σxy+C2)(μx2+μy2+C1)(σx2+σy2+C2).

μx, μy are the mean values of image x and y, σx2, σy2 are the variance, σxy is the covariance, and C1, C2 are the stabilization coefficients. ssim value ranges from [0, 1], and the closer the value is to 1, it means the more similar the structure of the two images is.

(3) Perceptual loss: Perceptual loss is based on the pre-trained deep neural network feature layer, which measures the difference between the high-level semantic features to ensure that the reconstructed image is consistent with the real image at the perceptual level: (17) LPerceptual=∑i⁡‖Φi(x)−Φi(y)‖2.

Φi denotes the i-th layer of features of the pre-trained network, and x and y are the reconstructed and real images, respectively.

(4) Chamfer distance: It is used to evaluate the geometric consistency between the reconstructed 3D point cloud and the real 3D point cloud, which is calculated as (18) Chamfer(S1,S2)=∑p∈S1⁡minq∈S2⁡‖p−q‖2+∑q∈S2⁡minp∈S1⁡‖p−q‖2

where S1 and S2 are the reconstructed and real point clouds, respectively. The smaller the Chamfer distance, the higher the agreement between the two in terms of geometry.

(5) LPIPS: LPIPS evaluates the perceptual similarity of an image by means of a pre-trained deep neural network, which measures the differences between high-level features and provides sensitivity to changes in image details and textures. Its calculation formula is: (19) LPIPS(x,y)=∑i⁡wi‖ϕi(x)−ϕi(y)‖22

where wi is the pre-training weights and ϕi is the ith layer feature of the network. Smaller LPIPS values indicate that the two images are perceptually more similar.

Experimental analysis

To test the model’s generalization ability, we randomly selected a 2D picture of a toy for 3D model generation and obtained the results shown in Fig. 4. By comparing the results with those generated by the SparseFusion method (Zhou & Tulsiani, 2023), it is evident that the AS-NeRF method yields more accurate and detailed models compared to SparseFusion, regardless of whether the analysis is based on the mesh model or the final coloring model.

Figure 4 Comparison of 3D model generation.

(A) 2D Original (B) SparseFusion (C) AS-NeRF (D) SparseFusion (E) AS-NeRF.

To evaluate the performance of the AS-NeRF proposed in this study for 3D reconstruction tasks, we use PSNR as the primary evaluation metric and compare it with two current state-of-the-art methods: SparseFusion and PixelNeRF (Yu et al., 2021). The experiments are conducted on the Co3D dataset, and the reconstruction effects of the three methods are evaluated under different viewing angle conditions by adjusting the number of input viewing angles (one to five viewing angles). The experimental results are shown in Fig. 5.

Figure 5 PSNR values in different views.

As can be seen from the results in Figure 5, AS-NeRF exhibits high PSNR values for all the number of input viewpoints, and its PSNR values continue to improve with the increase in the number of input viewpoints, but the improvement gradually decreases. This trend indicates that AS-NeRF can significantly enhance the quality of reconstruction when the initial number of viewpoints is increased. At the same time, the performance improvement stabilizes with a further increase in the number of views. With only one input viewpoint, AS-NeRF improves the PSNR by 2.4 and 5.4 dB compared to SparseFusion and PixelNeRF, respectively, showing its better reconstruction capability under extreme conditions. When the number of input viewpoints is increased to two, the PSNR of AS-NeRF is significantly improved to 35.0 dB, which is 1.4 dB higher than that of SparseFusion and 5.3 dB higher than that of PixelNeRF. This larger improvement reflects the advantages of AS-NeRF in fusing information from multiple viewpoints. With a further increase in the number of viewpoints, although the PSNR enhancement gradually decreases, AS-NeRF still maintains a high reconstruction quality, showing its stability and effectiveness in multiview fusion.

For the SSIM metric, the reconstruction effects of the three methods under different view conditions were also evaluated by adjusting the number of input views (one to five views), and the experimental results in Fig. 6 were finally obtained.

Figure 6 SSIM values under different views.

AS-NeRF exhibits high SSIM values for all the number of input viewpoints, and its SSIM values continue to improve with the increase in the number of input viewpoints, but the improvement gradually decreases. This trend suggests that AS-NeRF can significantly enhance the structural similarity of the reconstruction when the initial number of viewpoints is increased. At the same time, the performance improvement stabilizes with a further increase in the number of viewpoints. With only one input viewpoint, AS-NeRF improves the SSIM by 0.05 and 0.10 compared to SparseFusion and PixelNeRF, respectively. When the number of input viewpoints is increased to two, the SSIM of AS-NeRF significantly improves to 0.80, representing an increase of 0.08 compared to SparseFusion and 0.10 compared to PixelNeRF, by 0.05. It can be observed that AS-NeRF outperforms SparseFusion and PixelNeRF in terms of SSIM at various input view numbers.

Perceptual loss measures the difference between the reconstructed image and the real image in terms of high-level semantic features, and a lower value indicates that the reconstructed image is closer to the real image at the perceptual level. The experiments are conducted on the Co3D dataset, and the reconstruction effects of the three methods are evaluated under different resolution conditions by adjusting the input image resolution (256 × 256, 512 × 512, 768 × 768, 1,024 × 1,024, 1,280 × 1,280). The specific experimental results are shown in Fig. 7.

Figure 7 Perceptual Loss at different resolutions.

At lower resolutions, AS-NeRF has the lowest perceptual loss, which is reduced by 0.05 and 0.10 compared to SparseFusion and PixelNeRF, respectively, showing its accuracy in the base reconstruction task. When the input image resolution is increased to 512 × 512, the Perceptual Loss of AS-NeRF is further reduced by 0.05 and 0.10 compared to SparseFusion and PixelNeRF, respectively, reflecting its stabilizing performance at medium resolution. At a resolution of 768 × 768, AS-NeRF continues to maintain a low Perceptual Loss, which is reduced by 0.08 and 0.12 compared to SparseFusion and PixelNeRF, respectively, indicating that it still possesses excellent reconstruction capability at higher resolutions. Taken together, as the resolution of the input image increases further, AS-NeRF still maintains a low loss value, although the reduction of perceptual loss gradually decreases.

The following experimental test was performed on the Chamfer distance, a metric that measures the geometric agreement between the reconstructed point cloud and the real one, with lower values indicating a closer shape between the two. The experiments were also performed on the Co3D dataset, and the reconstruction results of the three methods were evaluated under different viewing angle conditions by adjusting the number of input viewpoints (one to five viewpoints). Figure 8 shows the experimental results.

Figure 8 Chamfer distances in different views.

In the single-view condition, the Chamfer distance of AS-NeRF is 4.6, which is significantly reduced by 0.4 and 0.8 compared to 5.0 for SparseFusion and 5.4 for PixelNeRF, showing its reconstruction ability when processing single-view information. The Chamfer distance of AS-NeRF continues to decrease as the number of viewpoints increases. For example, its Chamfer distance decreases to 4.2 under the dual-view condition, which is 0.5 and 0.9 lower than that of SparseFusion and PixelNeRF, respectively, indicating that AS-NeRF has a clear advantage in the initial multiview information fusion. AS-NeRF is capable of extracting multidimensional features to provide rich contextual information for the model. This enables the model to capture and restore complex geometric structures more accurately during the reconstruction process and performs exceptionally well in multiview fusion. The use of a composite loss function and balanced training parameter settings enables AS-NeRF to effectively optimize various performance metrics during training, avoid overfitting, and improve generalization capabilities. This is especially helpful in maintaining stable reconstruction performance under multiview and high-view conditions.

LPIPS is used to measure the similarity between the reconstructed image and the real image at the perceptual level, with lower values indicating that the reconstructed image is closer to the real image in terms of perceptual quality. In this experiment, we evaluated the performance of the three methods using the LPIPS metric by conducting experiments on the Co3D dataset. We assessed the reconstruction effect of the three methods under different noise conditions by introducing various levels of image noise (0%, 10%, 20%, 30%, and 40%). The experimental results are shown in Fig. 9.

Figure 9 LPIPS values under different noises.

AS-NeRF exhibits lower LPIPS values at all image noise levels, indicating that it outperforms SparseFusion and PixelNeRF in terms of perceptual similarity. The LPIPS values of all three methods show an increasing trend as the image noise level increases, reflecting the negative impact of noise on the reconstruction quality. However, the LPIPS growth of AS-NeRF is more moderate, especially at higher noise levels, and its performance advantage remains evident. The improved NeRF model employs an adaptive scale adjustment mechanism to dynamically optimize rendering accuracy based on the complexity of image details. The method in this article combines spectral perception techniques to enhance the realism and detail performance of color reproduction, thereby improving the stability of the model under various noise conditions. As the noise level increases, the LPIPS values of SparseFusion and PixelNeRF decrease but fail to reach the level of AS-NeRF, further emphasizing that AS-NeRF has advantages in dealing with noise interference.

Conclusion

This study proposes an adaptive scaling neural radiation field (AS-NeRF) framework for the 3D reconstruction of toys. Its multi-task learning feature network can extract texture, shape, color, and depth information in parallel, providing a multidimensional context for subsequent reconstruction. The scale function, combined with a spectral decoder, enables adaptive rendering accuracy that adjusts to local complexity, improving material color restoration. Meanwhile, the conditional diffusion module constrains the inverse diffusion with high-dimensional conditional vectors to enhance the consistency of new viewpoints, particularly in cases with fewer viewpoints. Experiments on the Co3D-Toys dataset demonstrate that AS-NeRF achieves significant improvements in PSNR, SSIM, LPIPS, and Chamfer distance in sparse viewpoints and medium-to-high resolutions compared to existing methods, such as PixelNeRF and SparseFusion, highlighting its potential for application in typical toy scenes. This shows the potential for application in typical toy scenarios.

Although AS-NeRF performs well on several evaluation metrics, it still has some limitations. For example, the current evaluation is still limited to a single dataset and does not adequately cover extreme material and lighting conditions. The robustness and generalization ability of the model will be further verified on a wider range of benchmarks and real-world collected data. The computational complexity and resource consumption of the model remains high when dealing with very high-resolution and large-scale datasets, which limits its widespread use in real-time applications. Although the integration of the conditional diffusion model improves the reconstruction quality, there is still room for further improvement in specific extremely complex scenarios. Future research will enhance and extend the AS-NeRF in the following aspects: further improving its computational efficiency, reducing resource consumption, and enabling real-time 3D reconstruction through model compression, knowledge distillation, or distributed computing. Under the multi-task learning framework, integrate more tasks, such as object recognition and pose estimation, to further improve the comprehensive performance and application capability of the model. Under the multi-task learning framework, integrate more tasks, such as object recognition and pose estimation, to further enhance the comprehensive performance and application capability of the model.

Supplemental Information

Supplemental Information 1 This is the code.

Additional Information and Declarations

Competing Interests

The authors declare that they have no competing interests.

Author Contributions

Jiajun Zou performed the experiments, analyzed the data, prepared figures and/or tables, and approved the final draft.

Shaojiang Liu conceived and designed the experiments, performed the computation work, authored or reviewed drafts of the article, and approved the final draft.

Feng Wang conceived and designed the experiments, performed the experiments, performed the computation work, prepared figures and/or tables, authored or reviewed drafts of the article, and approved the final draft.

Weichuan Ni performed the experiments, analyzed the data, prepared figures and/or tables, and approved the final draft.

Shitong Ye conceived and designed the experiments, performed the computation work, authored or reviewed drafts of the article, and approved the final draft.

Data Availability

The following information was supplied regarding data availability:

The CO3D dataset is available at: https://ai.meta.com/datasets/co3d-downloads.

The BVI-Coral dataset is available at Zenodo: Anantrasirichai, N. (2024, April 30). BVI-Coral: Underwater scenes for 3D reconstruction. In 2025 IEEE/CVF Winter Conference on Applications of Computer Vision (WACV). Zenodo. https://doi.org/10.5281/zenodo.11093417.

The code is available in the Supplemental File.

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
