# Peer review of "D reconstruction of toys based on adaptive scaled neural radiation field"

_PeerJ Computer Science, doi:10.7717/peerj-cs.3053_

## Round 0.1 · original submission · Major Revisions

Dear authors

Thanks for your submission, after carefully considering your article and input received from the experts in the field. I would like to inform you that your manuscript is being recommended for major changes to be addressed as mentioned by the experts and mine observations below.

Please carefully revise and update your manuscript in light of these suggestions and resubmit for a second round of opinion.

Academic Editor comments:

Abstract: please consicely revise the abstract to present a clear picture of the study

Novelty: justify the contribution in introduction section

Practical needs: Please briefly explain the practical implications of the research and benefits to the community

Language: technical language needs a thorough overhaul to clearly describe the study for potential readers.

Thank you

**Language Note:** The review process has identified that the English language must be improved. PeerJ can provide language editing services - please contact us at [email protected] for pricing (be sure to provide your manuscript number and title). Alternatively, you should make your own arrangements to improve the language quality and provide details in your response letter. – PeerJ Staff

·

Basic reporting

he manuscript introduces AS-NeRF for toy 3D reconstruction and is well-structured with relevant references. However, the writing suffers from frequent grammatical errors, inconsistent terminology (e.g., “bulk density” vs. “scene density”), and redundancy in the abstract and introduction. Some model components (e.g., spectral decoder, conditional diffusion integration) are insufficiently detailed for replication. Authors should improve clarity, provide training code or logs, and standardize equations and visualizations to meet publication standards.

Experimental design

No ablation study is provided to isolate the contributions of the multi-task network, adaptive scaling, or spectral decoder. The experiments compare to only two baselines and lack statistical significance testing. Despite using Co3D, class diversity, sample size per category, and data preprocessing steps are not reported. The study would benefit from greater methodological transparency, clearer reproducibility documentation, and stronger comparative benchmarking.

Validity of the findings

Data access is limited to the Co3D dataset; trained models, code, and sample outputs are not shared. The conclusions overstate robustness without sufficient evidence from diverse scenes or rigorous evaluation protocols.

Additional comments

1. Lack of Scientific Rigor and Reproducibility
The methodology is overly reliant on vague pseudo-code and abstract equations without implementation details, making replication virtually impossible. Critical training parameters (e.g., diffusion steps, spectral decoder architecture, loss weight tuning) are either glossed over or buried in verbose, mathematically incorrect notation. No code or pretrained models are provided, severely limiting reproducibility and transparency.

2. Inflated Claims Without Statistical Support
The authors claim superior performance based on PSNR, SSIM, LPIPS, etc., yet fail to report standard deviations, confidence intervals, or statistical significance testing. The absence of proper benchmarking against more than two outdated baselines (SparseFusion, PixelNeRF) further undermines the credibility of their performance claims.

3. Poor Writing and Technical Communication
The manuscript is riddled with grammatical errors, repetitive phrasing, and inconsistent terminology (e.g., "scene density" vs. "bulk density"), making it exhausting to read. Many equations include typographic artifacts (e.g., “ÿ8”, “ý”) and the figures are low-resolution, unlabeled, and inadequately described, which fails basic standards of scholarly presentation.

4. Conceptual Overload with No Empirical Isolation
The paper attempts to combine multi-task learning, adaptive scaling, spectral rendering, and conditional diffusion—all at once—without performing any ablation or sensitivity analysis. This design-by-aggregation approach dilutes scientific value, as it's impossible to assess the contribution or necessity of each module in the final outcome.

Reviewer 2 ·

Basic reporting

1) The authors have outlined the research questions and experimental results in the abstract, but the description of the specific methodological details of AS-NeRF, such as multi-task feature extraction and the specific role of conditional diffusion modeling, is rather brief. It is suggested that a brief description of these key technical points be appropriately supplemented in the abstract, so as to enable the readers to have a clearer understanding of the technical contributions of this paper.
2) Although the introduction section presents the background of the study, it is recommended that the authors further clearly present the link between the motivation for the study and the existing research gaps, and specifically articulate how the technical means of this paper can overcome the shortcomings of the existing methodology, so as to better reflect the necessity of the research in this paper.
3) The authors list a number of related research results, but the advantages and disadvantages of these studies, the similarities and differences of technical routes, and the relationship between the methods lack in-depth and systematic comparative analysis, and it is recommended to supplement the comprehensive comparison between the literature to help readers better understand the technological development of the field of the vein.

Experimental design

4) The overall framework of AS-NeRF shown in Figure 1 is only briefly demonstrated, and it is recommended that the authors describe more clearly the relationship of data flow between different modules, especially the specific interaction details between MTL-FeatureNet, conditional diffusion model and NeRF model.
5) The authors used the Co3D dataset for their experiments, but information on the number of samples in the dataset, the class distribution of the data, the resolution of the images, and the viewpoint settings were not described in detail.
6) The rationale for choosing SparseFusion and PixelNeRF as comparison methods is not adequately addressed, and it is recommended that further clarification be provided on the specific details of the dataset as well as the specific reasons for the choice of comparison methods.

Validity of the findings

7) Although Figure 3 in the paper shows the implementation process of the improved NeRF model fusing adaptive scale adjustment and spectral sensing technology, the illustration is too simple and does not fully reflect how the adaptive scale mechanism and spectral sensing can be combined to play a role. It is suggested that the authors should further optimize the detailed performance of this figure and strengthen the clear explanation of the various parts of the illustration in the main text.

Additional comments

I recommend for (Major Corrections)

Annotated reviews are not available for download in order to protect the identity of reviewers who chose to remain anonymous.
Cite this review as

---

## Round 0.2 · accepted · Accept

Dear authors

We are happy to let you know that your manuscript has been accepted for publication following a successful peer review and revision process.

The manuscript will now proceed to the production stage. You will receive a galley proof for final review before publication. Kindly ensure all author details are accurate and respond promptly to any queries from the production team.

Congratulations on your acceptance, and thank you for choosing our Journal as the venue for your work. We look forward to your continued contributions to the scientific community.

·

Basic reporting

Authors addressed my comments.

Experimental design

Authors addressed my comments.

Validity of the findings

Authors addressed my comments.

Additional comments

Authors addressed my comments.

Reviewer 2 ·

Basic reporting

Accepted

Experimental design

Accepted

Validity of the findings

Accepted

Cite this review as